# Estrogen receptor alpha somatic mutations Y537S and D538G confer breast cancer endocrine resistance by stabilizing the activating function-2 binding conformation

Sean W Fanning[1†], Christopher G Mayne[2,3,4†], Venkatasubramanian Dharmarajan[5†], Kathryn E Carlson[3], Teresa A Martin[3], Scott J Novick[5], Weiyi Toy[6], Bradley Green[1], Srinivas Panchamukhi[1], Benita S Katzenellenbogen[7], Emad Tajkhorshid[2,4], Patrick R Griffin[5], Yang Shen[8], Sarat Chandarlapaty[6], John A Katzenellenbogen[3], Geoffrey L Greene[1*]

[1]Ben May Department for Cancer Research, University of Chicago, Chicago, United States; [2]Beckman Institute for Advanced Science and Technology, University of Illinois at Urbana-Champaign, Urbana, United States; [3]Department of Chemistry, University of Illinois at Urbana-Champaign, Urbana, United States; [4]Department of Biochemistry, Center for Biophysics and Computational Biology, University of Illinois at Urbana-Champaign, Urbana, United States; [5]Department of Molecular Therapeutics, The Scripps Research Institute, Jupiter, United States; [6]Human Oncology and Pathogenesis Program, Memorial Sloan Kettering Cancer Center, New York, United States; [7]Department of Molecular and Integrative Physiology, University of Illinois Urbana-Champaign, Urbana, United States; [8]Department of Electrical and Computer Engineering, TEES-AgriLife Center for Bioinformatics and Genomic Systems Engineering, Texas A&M University, College Station, United States

*For correspondence: ggreene@uchicago.edu

†These authors contributed equally to this work

Competing interests: The author declares that no competing interests exist.

**Abstract** Somatic mutations in the estrogen receptor alpha (ERα) gene (*ESR1*), especially Y537S and D538G, have been linked to acquired resistance to endocrine therapies. Cell-based studies demonstrated that these mutants confer ERα constitutive activity and antiestrogen resistance and suggest that ligand-binding domain dysfunction leads to endocrine therapy resistance. Here, we integrate biophysical and structural biology data to reveal how these mutations lead to a constitutively active and antiestrogen-resistant ERα. We show that these mutant ERs recruit coactivator in the absence of hormone while their affinities for estrogen agonist (estradiol) and antagonist (4-hydroxytamoxifen) are reduced. Further, they confer antiestrogen resistance by altering the conformational dynamics of the loop connecting Helix 11 and Helix 12 in the ligand-binding domain of ERα, which leads to a stabilized agonist state and an altered antagonist state that resists inhibition.
DOI: https://doi.org/10.7554/eLife.12792.001

## Introduction

The estrogen receptor α (ERα) is a ligand-activated nuclear hormone receptor and a major regulator of cell growth, survival, and metastasis in a large fraction of breast cancers. Inhibiting the action of

**eLife digest** Around one in every eight women will be diagnosed with breast cancer in their lifetime. Hormone-based therapies – also referred to antiestrogen drugs – target a protein called estrogen receptor alpha and are effective treatments for the majority of these cancers. Unfortunately, about half of patients will develop recurrent breast cancers even though the cancer continues to produce the target of the drugs.

The estrogen receptor alpha drives breast cancer in a number of ways, many of which require the receptor to be activated by binding to the hormone estrogen. When estrogen binds it causes the receptor to change shape to expose a surface where other proteins called coactivators can bind. Once a coactivator is bound, the estrogen receptor is active and signals the cancer cell to grow, divide, invade local tissues, and spread to new sites in the body.

Antiestrogen drugs competitively block the binding of estrogen to the receptor and cause the receptor to take on a different shape that inhibits the binding of the coactivator. However, recent studies identified mutations at specific sites in the gene that encodes estrogen receptor alpha in a large subset of patients with breast cancers that have spread. These mutations make the receptor resistant to antiestrogen drugs, and two mutations (called Y537S and D538G) account for approximately 70% of cases. However, it was not clear how these mutations altered the activity of estrogen receptor alpha at the molecular level.

Fanning, Mayne, Dharmarajan et al. now show these two most common mutations allow estrogen receptor alpha to bind to the coactivator in the absence of hormone. This unfortunately also reduces the effectiveness of one of the mostly widely administered antiestrogen therapies – a drug called tamoxifen. However, Fanning, Mayne, Dharmarajan et al. also show that the newer and more potent antiestrogens that are currently under examination in clinical trials should be highly effective at treating the cancers with the mutated versions of estrogen receptor alpha.

Applying the knowledge gained from these new findings toward the development of new antiestrogens could help reverse the impact of these common mutations. If successful, these new drugs will provide life-saving treatments for many breast cancer patients.
DOI: https://doi.org/10.7554/eLife.12792.002

ER$\alpha$ with selective estrogen receptor modulators (SERMs) or selective estrogen receptor degraders (SERDs), or reducing endogenous estrogen levels with aromatase inhibitors (AI), are effective treatments for many of these breast cancers (*Strasser-Weippl and Goss, 2005*). Due to their efficacy and broad therapeutic indices, antiestrogens can be administered sequentially for progressive disease over the course of several years (*Toy et al., 2013*). Unfortunately, despite continued expression of ER$\alpha$, the majority of metastatic breast cancers that initially respond to endocrine therapies become refractory.

Recently, somatic mutations in the ER$\alpha$ gene (*ESR1*) were linked to acquired resistance to endocrine therapies of breast cancer (*Toy et al., 2013*; *Merenbakh-Lamin et al., 2013*; *Robinson et al., 2013*; *Li et al., 2013*; *Jeselsohn et al., 2014*). Approximately 25% of patients who previously received SERM/SERD/AI treatments for an average of five years presented with conserved somatic mutations that were not identified in primary (untreated) tumors. The most prevalent ER$\alpha$ point mutations were Y537S and D538G, while several others were identified at significantly reduced frequencies. Importantly, breast cancer cell-based studies revealed that the Y537S and D538G mutations conferred hormone-independent activation of ER$\alpha$ and reduced the inhibitory potency and efficacy of clinically prescribed SERMs and SERDs (*Toy et al., 2013*; *Merenbakh-Lamin et al., 2013*; *Robinson et al., 2013*; *Li et al., 2013*; *Jeselsohn et al., 2014*; *Carlson et al., 1997*). Notably, the constitutive activity and antagonist resistance of the Y537S and E380Q mutations were first described in cell models in 1996 (*Weis et al., 1996*), and shortly thereafter, the Y537N mutation was found in a clinical sample of metastatic breast cancer (*Zhang et al., 1997*). However, no clinical follow-up studies were reported until 2013.

The Y537S and D538G mutations are especially interesting because they occur at the N-terminus of Helix 12 (H12) in the ER$\alpha$ ligand-binding domain (LBD). Structurally, ER$\alpha$ LBD is an $\alpha$-helical bundle, with the C-terminal helix, H12, functioning as a key structural component of the activating

function-2 (AF-2) cleft that governs the agonist or antagonist state of the receptor. In the agonist conformation (e.g. estradiol (E2)-bound), H12 covers the ligand-binding pocket, docking between Helices 3 (H3) and 11 (H11), thereby facilitating coactivator recruitment to the AF-2 cleft via canonical LXXLL coactivator sequence motifs. In contrast, in the antagonist state (e.g.SERM-bound), H12 occupies the AF-2 cleft using its own LXXML sequence, thereby blocking coactivator recruitment and ERα action.

In this study, biophysical assays reveal the impact of the Y537S and D538G mutations on ERα LBD ligand and co-regulator binding affinity. Additionally, x-ray crystal structures and atomistic molecular dynamics (MD) simulations uncover altered conformations adopted by the mutant receptors in the absence and presence of agonists and antagonists. Together, these findings present a molecular explanation for how the Y537S and D538G mutations elevate the basal or constitutive activity of ERα and confer resistance to the beneficial effects of the SERM, SERD, and AI therapies. A comprehensive understanding of how these and other gain-of-function mutations alter the structure and function of ERα is crucial to development of more efficacious and potent inhibitors to target these mutant receptors in the clinic.

## Results

### Y537S and D538G promote constitutive coactivator binding to ERα

An established time-resolved Förster Resonance Energy Transfer (tr-FRET) assay that determines the affinity of the steroid receptor coactivator 3 nuclear receptor domain (SRC3 NRD) for the ERs was used to investigate differences among the WT, Y537S, and D538G (*Tamrazi et al., 2005*, *Jeyakumar et al., 2011*). SRC3 was chosen because of its abundance in breast cancer cells and high affinity for ERα (*Liao et al., 2002*). *Table 1* summarizes all SRC3 coactivator binding affinities. SRC3 NRD bound to the E2-saturated WT ERα LBD with high affinity ($K_d$ = 2.67 ± 0.5 nM) while no binding was detected in the absence of E2 or in the presence of the SERM 4-hydroxytamoxifen (TOT; the active metabolite of tamoxifen) (*Figure 1*). In contrast, the SRC3 NRD bound to Y537S and D538G ERα in the absence of E2, with affinities of 13.6 ± 2.0 nM and 151 ± 20 nM, respectively, and the binding curves reached approximately 60% of the maximum (*Figure 1*). When Y537S and D538G were pre-saturated with E2, the SRC3 binding curves reached the same maximum as WT with E2, with the coactivator binding affinity for the mutants being comparable or slightly higher than WT (WT $EC_{50}$ = 2.67 ± 0.5 nM; Y537S = 0.59 ± 0.1 nM; D538G = 3.65 ± 0.4 nM) (*Figure 1*). Neither the

WT nor the mutants bound coactivator when pre-incubated with saturating concentrations TOT (*Figure 1*).

To determine the potency of ligands to affect coactivator binding to the ER, ligand was titrated into a constant amount of SRC3 and ER and measured by tr-FRET. Addition of E2 resulted in increased coactivator affinity to the Y537S ($EC_{50}$ = 1.6 ± 1.2 nM) and D538G ($EC_{50}$ = 2.2 ± 0.1 nM) ERα LBD. Interestingly, the $EC_{50}$ value was somewhat reduced for WT ($EC_{50}$ = 13.8 ± 0.9 nM) (*Figure 1—figure supplement 1*). TOT abolished basal Y537S and D538G SRC3 binding in the absence of agonist. To mimic this reversal in WT, which does not bind SRC3 NRD without ligand, a low concentration of E2 was added to WT-ER to recruit SRC3 NRD to about 50% of maximal (data not shown). As expected, titration of TOT reversed the binding of SRC3 NRD by the mutant ER and E2-primed WT. The $EC_{50}$ values for suppressing SRC3 binding of the mutant (done in the absence of agonist) were comparable to the $K_i$ values for WT. The $K_i$ of TOT was 1.82 ± 0.30 nM

**Table 1.** SRC3 NRD and ligand recruitment affinities for the WT and mutant ERα LBDs. LBD, ligand-binding domain.

|  | SRC-3 NRD $K_d$ (nM) |
|---|---|
| WT apo | No Recruitment |
| Y537S apo | 13.6 ± 2.0 |
| D538G apo | 151 ± 20 |
| WT-E2 | 2.67 ± 0.5 |
| Y537S-E2 | 0.59 ± 0.1 |
| D538G-E2 | 3.65 ± 0.40 |
|  | E2 $EC_{50}$ (nM) |
| WT | 13.8 ± 0.9 |
| Y537S | 1.6 ± 1.2 |
| D538G | 2.2 ± 0.1 |
|  | TOT $K_i$ (nM) |
| WT | 1.82 ± 0.30 |
| Y537S | 6.7 ± 0.40 |
| D538G | 0.79 ± 0.04 |

DOI: https://doi.org/10.7554/eLife.12792.003

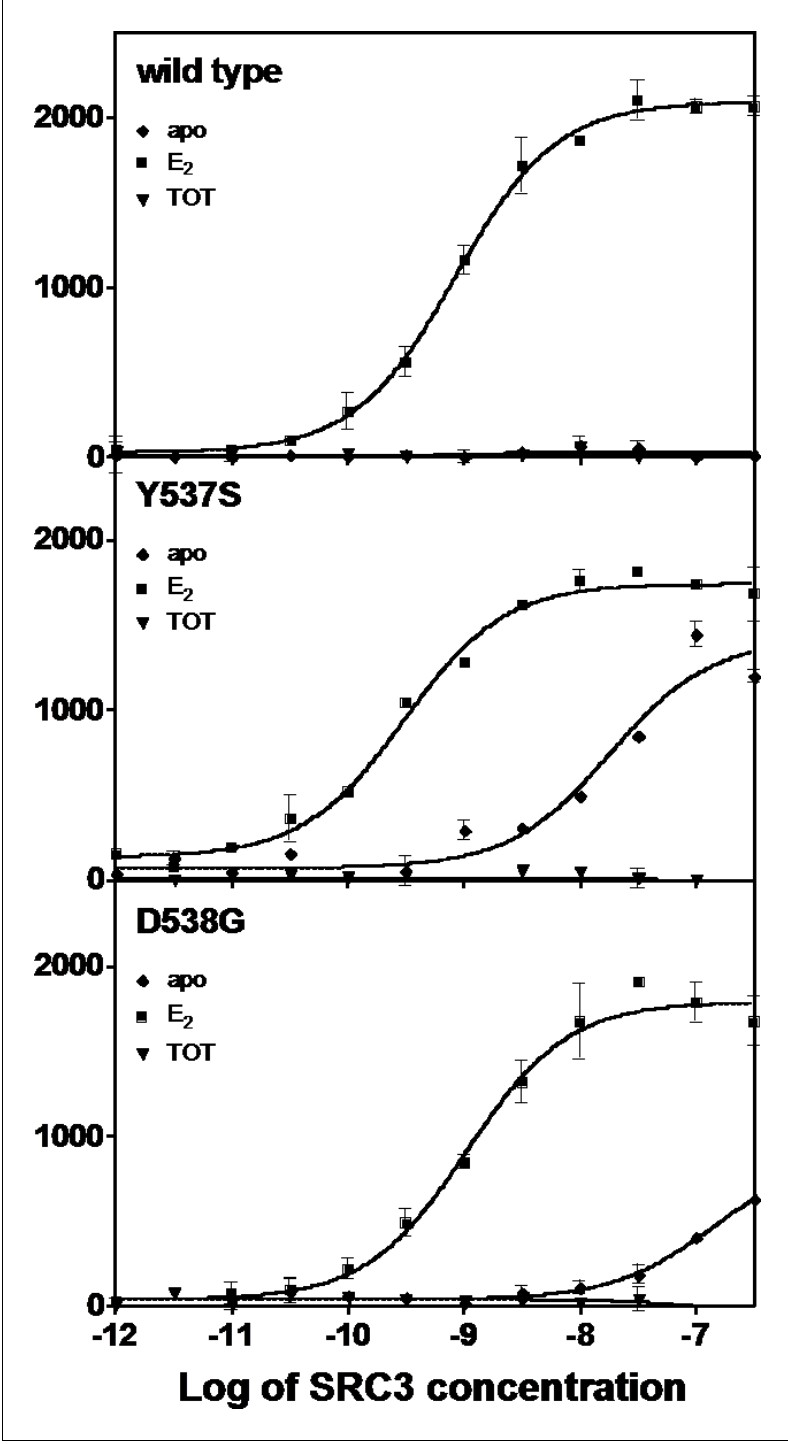

**Figure 1.** Binding of the SRC3 coactivator to WT, Y537S, or D538G ERα LBD in the absence or presence of E2 or TOT. LBD, ligand-binding domain.

DOI: https://doi.org/10.7554/eLife.12792.004

The following figure supplement is available for figure 1:

**Figure supplement 1.** Binding of the SRC3 coactivator to WT, Y537S, or D538G mutant ERα LBD with increasing concentrations of E2 or TOT.

DOI: https://doi.org/10.7554/eLife.12792.005

for WT, $6.7 \pm 0.40$ nM for Y537S, and $0.79 \pm 0.04$ nM for D538G.

## The Y537S and D538G mutants bind ligands with reduced affinity

Our earlier work demonstrated that SERMs were less potent in inhibiting the transcriptional activity of the ERα Y537S and D538G mutants compared to WT in breast cancer cells (*Toy et al., 2013*). The binding affinities of E2 with the WT and mutant ERα LBDs were measured using radioligand bindin-gligand-binding assays (*Carlson et al., 1997*). The affinity of E2 for WT-ER ($K_d = 0.26 \pm 0.13$ nM) is approximately five-fold greater than for the mutants, Y537S ($K_d = 1.43 \pm 0.55$ nM) and D538G ($K_d = 1.30 \pm 0.63$ nM) (*Figure 2*). *Table 2* summarizes all ligand-binding affinities for the WT and mutant ERα LBDs.

A competitive radioligand-binding assay with $^3$H-E2 as tracer was used to measure the relative competitive binding affinities (RBAs) of TOT for WT and the mutant-ERs (*Katzenellenbogen et al., 1973*; *Carlson et al., 1997*). The $K_i$ of TOT binding to WT was $0.337 \pm 0.018$ nM, whereas it was $2.61 \pm 0.60$ nM and $3.42 \pm 0.5$ nM for the Y537S and D538G mutants, respectively. Comparing the $K_i$ values, it is notable that the affinity of TOT for the Y537S and D538G mutants is impaired approximately 8- and 10-fold relative to WT (*Table 2*). This reduced binding affinity is consistent with the published lower inhibitory potency of TOT on the mutants in breast cancer cells (*Toy et al., 2013*). *Figure 3* shows representative radiometric competitive binding measurements.

## Biophysical basis for aberrant coregulator recruitment by Y537S and D538G ERα LBD mutants

### Proteolytic susceptibility

An established trypsin digestion assay was used to determine whether the conformational dynamics of the LBD H11-12 loop and H12 are altered as a result of the Y537S and D538G mutations (*Tamrazi et al., 2005*). The measured half-life for H11-12 loop and H12 cleavage ($t_{1/2}$) of the unli-ganded (*apo*) WT ERα LBD was 2 min, indicating that this region is highly mobile (*Figure 4A*). In contrast, the H11-12 loop and H12 region displayed significantly reduced proteolysis for *apo* D538G, with a $t_{1/2}$ of 19 min. A further reduction was observed for the H11-12 loop and H12 for *apo* Y537S with a $t_{1/2} = 87$ min. When incubated with saturating concentrations of E2, each of the LBDs displayed increased stability of the H11-12 loop and H12 with $t_{1/2} = 5$ min for the WT, 140 min for Y537S, and no detectable cleavage for D538G (*Figure 4A*). This lack of proteolysis for the D538G-E2

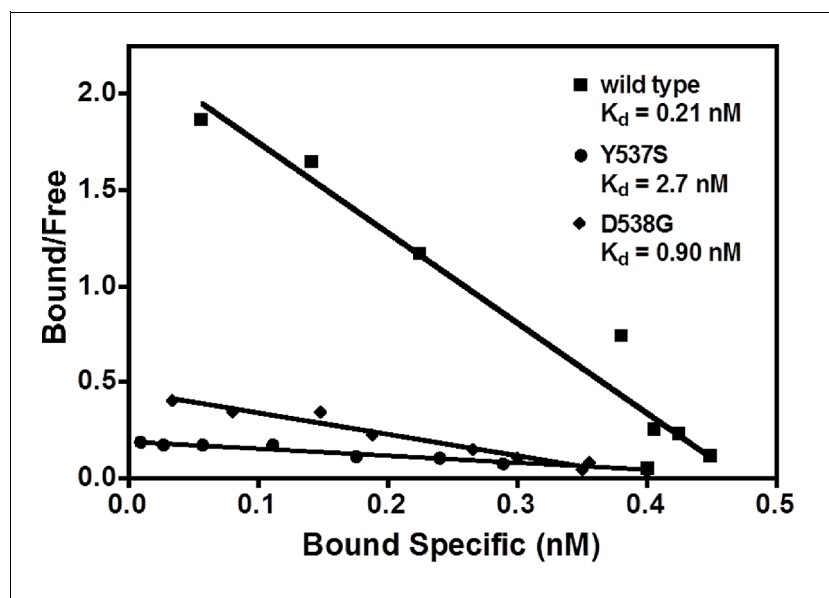

**Figure 2.** Determination of $K_d$ values of estradiol (E2) binding to wild type, Y537S, and D538G LBDs, by a direct binding assay. All slopes had an $r^2$ of 0.95 or better; shown is a representative experiment. For details, see Methods. LBD, ligand-binding domain.

DOI: https://doi.org/10.7554/eLife.12792.006

complex suggests that the H11-12 loop and H12 are stabilized and in a conformation that resists trypsin proteolysis. Importantly, the trend of H11-12 loop and H12 mobility observed for *apo* LBDs correlates with the relative coactivator binding affinities for *apo* WT and mutant LBDs as the Y537S mutant is the least dynamic and has the highest affinity for the coregulator.

## Hydrogen/deuterium exchange mass spectrometry

Hydrogen/deuterium exchange mass spectrometry (HDX-MS) was used to further dissect the con-sequences of Y537S and D538G ERα LBD

**Table 2.** Ligand-binding affinities.

|  | $K_d$ (nM) |
| --- | --- |
| WT-E2 | 0.26 ± 0.13 |
| Y537S-E2 | 1.43 ± 0.55 |
| D537G-E2 | 1.30 ± 0.63 |
|  | $K_i$ (nM) |
| WT-TOT | 0.337 ± 0.018 |
| Y537S-TOT | 2.61 ± 0.60 |
| D538G-TOT | 3.42 ± 0.50 |

DOI: https://doi.org/10.7554/eLife.12792.007

mutations on the conformational mobility of the H11-12 loop and H12. Perturbation in time-depen-dent deuterium uptake profiles (measured as protection to number of exchanged amide hydrogens with solvent deuterium between two states) is indicative of conformational alterations due to rear-rangement of amide hydrogen bonds (*Horn et al., 2006*). Differential amide HDX experiments were performed to compare the conformational dynamics of liganded and unliganded (*apo*) receptors. H11, the H11-12 loop, and H12 were all protected from solvent exchange for WT, D538G, and Y537S ERα LBD in the presence of E2 as compared to *apo* receptor (solvent exchange was lower for peptides containing these structural elements in the presence of ligand as compared to unliganded receptor), indicating the adoption of a more stable agonist-bound conformation matching that observed in x-ray co-crystal structures (*Figure 4B,C*, and *Figure 4—figure supplements 1–3*). For the unliganded states, the H12 of Y537S and D538G exhibited increased solvent exchange (depro-tection indicative of increased conformational dynamics) compared to WT ERα, suggesting that the mutant receptors adopt an alternative H12 conformation in the absence of E2. *Figure 4B–C* shows differential deuterium incorporation for the WT versus mutant ERα LBD in the *apo* states focusing on the H11-12 loop and H12 regions. *Figure 4—figure supplements 4* and 5 show the complete differ-ential HDX perturbation maps comparing the *apo* WT versus *apo* Y537S and D538G ERα LBD,

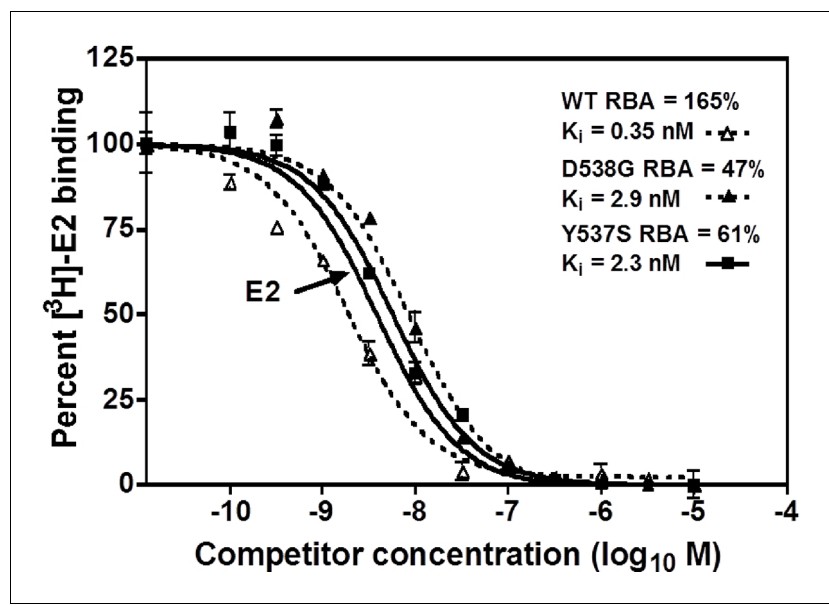

**Figure 3.** Relative binding affinity assay of wild type, Y537S, and D538G ligand-binding domains (LBDs), showing the TOT competition curves. With all proteins, the E2 curve is set to 100% and is shown only once. For details, see Methods.

DOI: https://doi.org/10.7554/eLife.12792.008

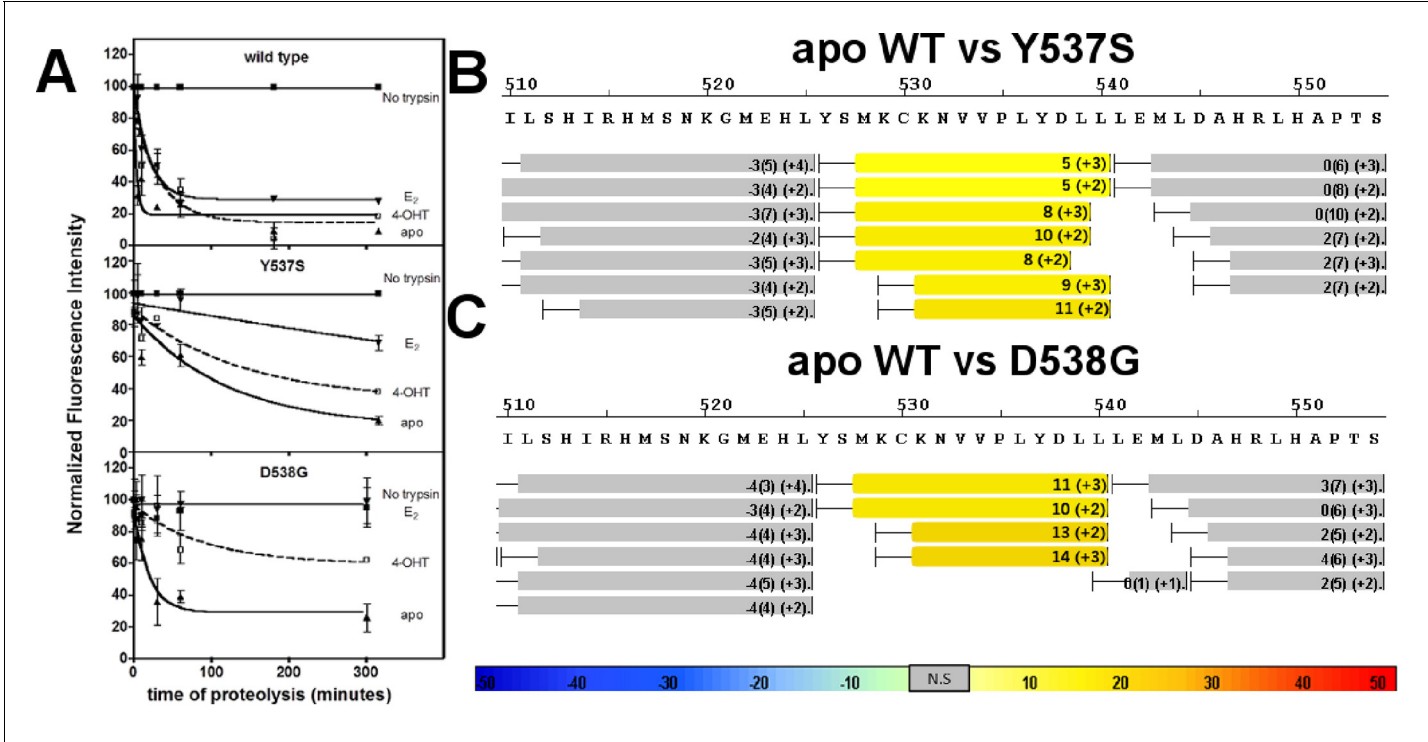

**Figure 4.** Conformational stability of WT and mutant ERα LBD H11-12 loop and H12. (**A**) Proteolytic susceptibility of the WT, Y537S, and D538G ERα LBD mutants in the *apo*, E2-bound, and TOT-bound states. (**B–C**) Deuterium uptake plot for the C-terminus of H11 along with the H11-12 loop and H12 for the *apo* WT vs Y537S ERα LBD (**B**), *apo* WT vs D538G ERα LBD (**C**). All HDX MS data represent an average of three replicates and are color coded from red to blue with warm colors representing increased conformational dynamics (red being the highest D2O uptake) and cool colors representing decreased conformational dynamics (blue being the lowest D2O uptake). All regions colored were determined to be statistically significant based on a paired two-tailed Students t-test. A legend is provided at the bottom. Grey indicates no statistically significant change between the two *apo* states. HDX, hydrogen/deuterium exchange; LBD, ligand-binding domain.

DOI: https://doi.org/10.7554/eLife.12792.009

The following figure supplements are available for figure 4:

**Figure supplement 1.** Complete differential amide HDX MS map of WT ERα LBD binding to E2.
DOI: https://doi.org/10.7554/eLife.12792.010

**Figure supplement 2.** Complete differential amide HDX MS map of Y537S ERα LBD mutant binding to E2.
DOI: https://doi.org/10.7554/eLife.12792.011

**Figure supplement 3.** Complete differential amide HDX MS map of D538G ERα LBD mutant binding to E2.
DOI: https://doi.org/10.7554/eLife.12792.012

**Figure supplement 4.** Complete differential HDX perturbation maps comparing the *apo* WT versus *apo* Y537S ERα LBD.
DOI: https://doi.org/10.7554/eLife.12792.013

**Figure supplement 5.** Complete differential HDX perturbation maps comparing the *apo* WT versus *apo* D538G ERα LBD.
DOI: https://doi.org/10.7554/eLife.12792.014

**Figure supplement 6.** Complete differential HDX perturbation map of WT ERα LBD with SRC3-NRD.
DOI: https://doi.org/10.7554/eLife.12792.015

**Figure supplement 7.** Complete differential HDX perturbation map of Y537S ERα LBD with SRC3-NRD.
DOI: https://doi.org/10.7554/eLife.12792.016

**Figure supplement 8.** Complete differential HDX perturbation map of D538G ERα LBD with SRC3-NRD.
DOI: https://doi.org/10.7554/eLife.12792.017

**Figure supplement 9.** Complete differential HDX perturbation map of WT ERα LBD with E2 and SRC3-NRD.
DOI: https://doi.org/10.7554/eLife.12792.018

**Figure supplement 10.** Complete differential HDX perturbation map of Y537S ERα LBD with E2 and SRC3-NRD.
DOI: https://doi.org/10.7554/eLife.12792.019

**Figure supplement 11.** Complete differential HDX perturbation map of D538G ERα LBD with E2 and SRC3-NRD.
DOI: https://doi.org/10.7554/eLife.12792.020

**Figure supplement 12.** *apo* Y537S x-ray crystal structure (Yellow) (PDB: 2B23) superimposed with WT-E2 complex structure (White) (PDB: 1GWR).

*Figure 4 continued on next page*

Figure 4 continued

DOI: https://doi.org/10.7554/eLife.12792.021

respectively. Interestingly, residues close in space to or within the AF-2 cleft (positions 310–325, 344–349, 370–380, and 405–410) of the *apo* Y537S also showed statistically significant increase in solvent exchange compared to *apo* WT. Similar deprotection was observed in residues 310–325 of the *apo* D538G. Together, the HDX data suggests that the Y537S and D538G mutants enables H12 to sample a suite of conformations that expose the AF-2 cleft at a greater frequency thereby facilitating coregulator recruitment in the absence of hormone. Furthermore, these data suggest that the Y537S mutant possesses a higher affinity for SRC3 as compared to D538G as it samples more frequently AF-2-cleft conformers that facilitate coregulator binding, in agreement with our in vitroSRC3 NRD-binding experiments.

In order to test our hypothesis that the increased deuterium uptake in the H12 region of the mutants was due to a rearrangement of amide hydrogen bonds that could facilitate coactivator recruitment, we performed differential HDX analysis for the WT, Y537S, and D538G ERα LBDs in the presence of SRC3 NRD, in the presence and absence of E2. Few statistically significant differences in solvent exchange were observed in the C-terminus of LBD when the WT ERα LBD was incubated with saturating concentrations of SRC3 NRD, with the exception of H11 (*Figure 4—figure supplement 6*). In contrast, the H11-12 loop showed statistically significant protection from exchange in the Y537S-SRC3 NRD complex, indicating that the region in the Y537S mutant was further stabilized by the inclusion of coregulator (*Figure 4—figure supplement 7*). These results suggest that H12 in the *apo* mutant receptor is in a more favorable conformation promoting co-activator binding when compared to *apo* WT. The magnitude of protection from solvent exchange observed in the AF-2 cleft in Y537S with SRC3 NRD was further increased upon addition of E2 indicating a more stable Y537S-SRC3-E2 complex (*Figure 4—figure supplement 10*). In contrast to Y537S, the H11-12 loop and H12 in the D538G mutant did not show a statistically significant difference in deuterium incorporation in the presence of SRC3 NRD alone, but did show increased protection from solvent exchange in these regions in the presence of E2 (*Figure 4—figure supplements 8* and *10*). This finding could be attributed to the low intrinsic SRC3 NRD-binding affinity of *apo* D538G as compared to Y537S (*Table 1*). Together, these data, along with the SRC3 NRD recruitment and trypsin susceptibility, suggest that the increased solvent exchange in H12 and AF-2 cleft residues for the *apo* Y537S is due to an altered conformation of H12 that promotes coactivator recruitment. This observation is apparent in the x-ray crystal structure of the *apo* Y537S. When compared to the WT-E2 complex (PDB: 1GWR), the serine at residue 537 in the *apo* Y537S (PDB: 2B23) replaces the phenolic side chain of WT Y537, exposing a solvent channel between the H11-12 loop and H3. Further, H12 is slightly displaced away from the ligand-binding pocket toward solvent (*Figure 4—figure supplement 12*). It is important to note that the HDX MS studies provide novel insight into the conformational mobility of the WT H12, in that this helix does not reach maximum structural stability until both hormone and coregulator are bound.

## Structural basis for H12 mutant hormone-independent activity

### X-ray crystallographic analysis of the D538G agonist states

High resolution x-ray crystal structures of the *apo* and agonist-bound states of the Y537S, obtained earlier, revealed near identical H12 conformations, in which S537 formed a hydrogen bond with D351 to adopt a stable agonist state in the absence of hormone (*Nettles et al., 2008*). In this study, we obtained x-ray crystal structures for the D538G mutant bound to E2, without added ligand (*apo*), and bound to a SERM (4-hydroxytamoxifen).

### D538G mutation induces pronounced conformational changes in the agonist-binding mode

The D538G-E2 complex structure was solved to 1.90 Å resolution by molecular replacement, with one dimer in the asymmetric unit (ASU). All crystallographic statistics are reported in *Table 3*. Overall, the structure presents a canonical ERα LBD-agonist binding state where H12 covers the ligand-binding pocket situated between H3 and H11, and the GRIP peptide occupies the AF-2 cleft. The E2

**Table 3.** Crystallographic data collection and refinement statistics.

| | ERα LBD D538G Apo | ERα LBD D538G-E2 | ERα LBD D538G-4OHT |
|---|---|---|---|
| **Data collection** | | | |
| Space group | $P2_1$ | $P2_1$ | $P2_12_12_1$ |
| a, b, c (Å) | 56.14, 82.66, 59.11 | 56.08, 84.18, 58.37 | 104.65, 104.65, 191.38 |
| α, β, γ (°) | 90.00, 111.05, 90.00 | 90.00, 108.83, 90.00 | 90.00, 90.00, 90.00 |
| Resolution range | 55.17-2.24 Å | 55.25-1.90 | 50.00-3.07 |
| **Number of reflections** | | | |
| (all/unique) | 91,607/24,107 | 169,519/40,361 | 60,232/9,874 |
| I/σ (highest resolution) | 2.37 | 2.36 | 1.70 |
| $R_{merge}$ | 11.4 | 7.3 | 11.4 |
| Completeness (%) | 98.9 | 99.3 | 96.7 |
| Redundancy | 3.8 | 4.2 | 6.1 |
| **Refinement** | | | |
| Rwork/Rfree | 19.8/24.9 | 17.9/21.4 | 21.6/28.3 |
| **No. Residues/chain** | | | |
| ERα LBD D538G | 241 | 242 | 216 |
| GRIP peptide | 6 | 6 | 0 |
| Water | 16 | 44 | 2 |
| Ligand | 0 | 1 | 1 |
| **RMSD** | | | |
| Bond lengths (Å) | 0.015 | 0.0170 | 0.0128 |
| Bond angles (°) | 1.76 | 1.5441 | 1.5356 |
| Chiral volume | 0.1117 | 0.1267 | 0.1036 |
| **Ramachandran plot statistics** | | | |
| Preferred number (%) | 428 (96.40%) | 443 (98.88%) | 1,563 (95.42%) |
| Additional allowed (%) | 3.60 (3.6%) | 5 (1.12% ) | 75 (4.58%) |
| Outliers (%) | 0 | 0 | 0 |

DOI: https://doi.org/10.7554/eLife.12792.022

ligand, GRIP peptide, and H12 (until residue L549) are well resolved in the map (*Figure 5—figure supplement 1*). No differences are observed in the residues comprising the ligand-binding pocket between the D538G-E2 and WT-E2 structures (*Gangloff et al., 2001*; *Eiler et al., 2001*; *Phillips et al., 2011*).

Pronounced conformational changes are observed in the loop connecting H11 and H12 (H11-12 loop, residues 531–537) in both monomers in the ASU for the D538G-E2 structure compared to the WT-E2 structure, although no appreciable changes are observed in most of H12. The H11-12 loop is displaced away from H3 and toward H11, accompanied by conformational changes in Y537 (*Figure 5*). In the WT-E2 structure, Y537 forms a hydrogen bond with N348 on H3, packing the H11-12 loop into the interior of the protein. In the D538G-E2 structure, however, the Y537 loses its hydrogen bond with N348, and its phenolic side chain is pointed toward bulk solvent. The space previously occupied by Y537 in WT is replaced by a well ordered water molecule in the mutant (observed in both monomers), which hydrogen bonds with the backbone amide of Y537 in between H3 and H12 (*Figure 5—figure supplement 2*). While the side-chain orientations are identical for residues 531–536 between both monomers in the ASU, the side chain of Y537 appears to adopt two different conformations, both facing solvent, while the main chain orientation of Y537 is identical in the two monomers. It should be noted that the phenolic oxygen of Y537 maintains the same hydrogen bond to N348 in every WT ERα LBD-agonist structure available in the PDB. Thus, this rotation of Y537 is unique to the D538G-E2 structure, and it brings the φ and ψ angles of residues 537 and 538 out of the α-helix region and into the allowed, more sheet-like region around -120° and 60° (defined by φ/

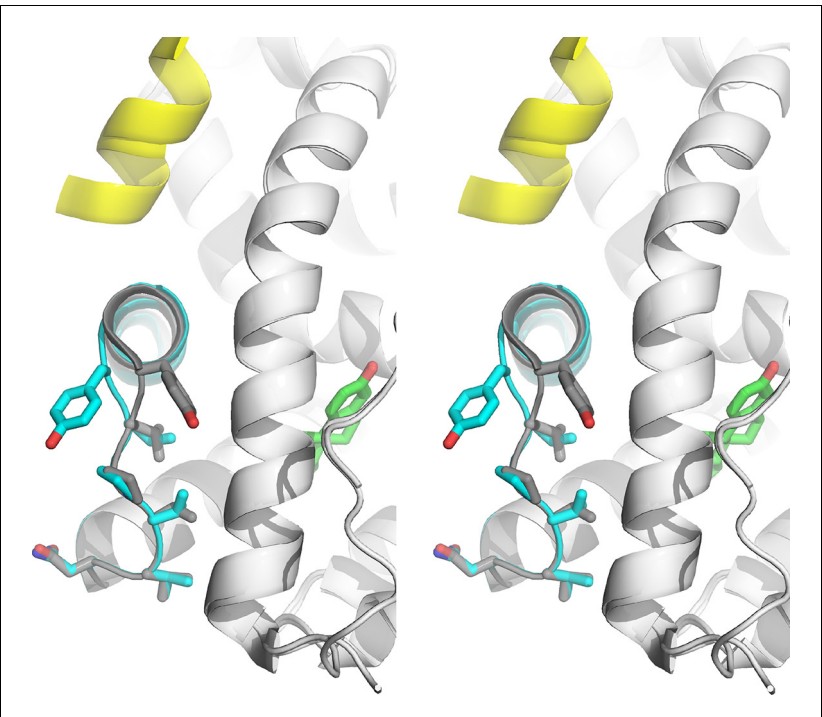

**Figure 5.** Stabilized D538G agonist state. Superposition stereo-view image of the residues comprising the H11-12 loop (531–537) of monomer A of the D538G-E2 (cyan) overlaid with monomer A of the WT-E2 structure (PDB: 1GWR). E2 is represented as green sticks. Coactivator peptide is shown as light-yellow ribbon.

DOI: https://doi.org/10.7554/eLife.12792.023

The following figure supplements are available for figure 5:

**Figure supplement 1.** Simulated annealing composite omit maps for the E2 (**A**) and TOT (**B**)-bound D538G ERα LBD crystal structures contoured to 1.5σ.

DOI: https://doi.org/10.7554/eLife.12792.024

**Figure supplement 2.** Y537 orientations for D538G and WT LBD.

DOI: https://doi.org/10.7554/eLife.12792.025

**Figure supplement 3.** Density of an unidentified small molecule in the ligand-binding site of the *apo* D538G x-ray crystal structure.

DOI: https://doi.org/10.7554/eLife.12792.026

---

ψ angle regions in the Ramachandran plot) (*Ramachandran et al., 1963*). In the resulting conformation, the α-helix of H12 begins at position 539 for the D538G-E2 structure rather than at 537 for the WT-agonist structures.

Few differences are observed between the unliganded and E2-bound D538G (C$_\alpha$ r.m.s.d. = 0.327 Å). The greatest conformational discrepancy between the two structures lies at Y537, which, in the unliganded structure, adopts a more WT-E2 like conformation by orienting toward H3 in chain A, thus returning the φ and ψ angles of residues 537 and 538 into the α-helical region. Y537 of chain B, however, matches the solvent-exposed orientation of the D538G-E2 structure whereby the φ and ψ angles for 537 and 538 are outside of the α-helical region. Based on this conformational asymmetry between the two monomers in *apo*-D538G ERα, Y537 can switch between the buried state observed in the WT-agonist structures and the solvent-exposed orientation of the D538G-E2 structure (*Figure 5—figure supplement 2A and B*). Thus, *apo* D538G has lost some—but not all—of the conformational attributes of the E2-bound mutant, which is consistent with its modest level of constitutive activity. Together, these structural features agree with our biophysical data showing that D538G can adopt an agonist state in the absence of hormone that recruits coregulator with a modest affinity.

It is of interest that the electron density map of *apo* D538G revealed some density in the ligand-binding pocket representing a non-specific small molecule likely acquired during the expression of

the protein, which remained during crystallization (*Figure 5—figure supplement 3*). A similar electron density was observed in the published *apo* Y537S (*Nettles et al., 2008*). The unidentified ligand is not of sufficient size to be a hormone nor is it near enough to H11 and H12 to interact with them. We believe that the unidentified small molecule in the ligand-binding site is an artifact of protein expression in bacteria, as reported earlier for the Y537S structure, and does not influence H11 and H12 nor the loop connecting them (*Nettles et al., 2008*).

## The dynamics of D538G-mediated alterations of the H11-12 loop

The previously published *apo* Y537S structure showed that S537 forms a hydrogen bond with D351 to adopt the agonist state in the absence of hormone thereby providing a clear conformational explanation for its constitutive activity (*Nettles et al., 2008*). In contrast, the *apo* D538G structure shows that this mutant may use a subtler mechanism to adopt the agonist conformation in the absence of hormone. In order to gain a better understanding of how the D538G mutation stabilizes the ERα LBD agonist conformation, MD simulations were performed on this mutant in the absence of ligand, and for WT ER (*Figure 6A*) in both the presence and absence of ligand. As was noted earlier, it has not been possible to obtain crystal structures of *apo* WT ER. Thus, to gain insights into the *apo* WT ERα LBD, MD simulations were performed by removing E2 from the ER complex prior to the dynamics run.

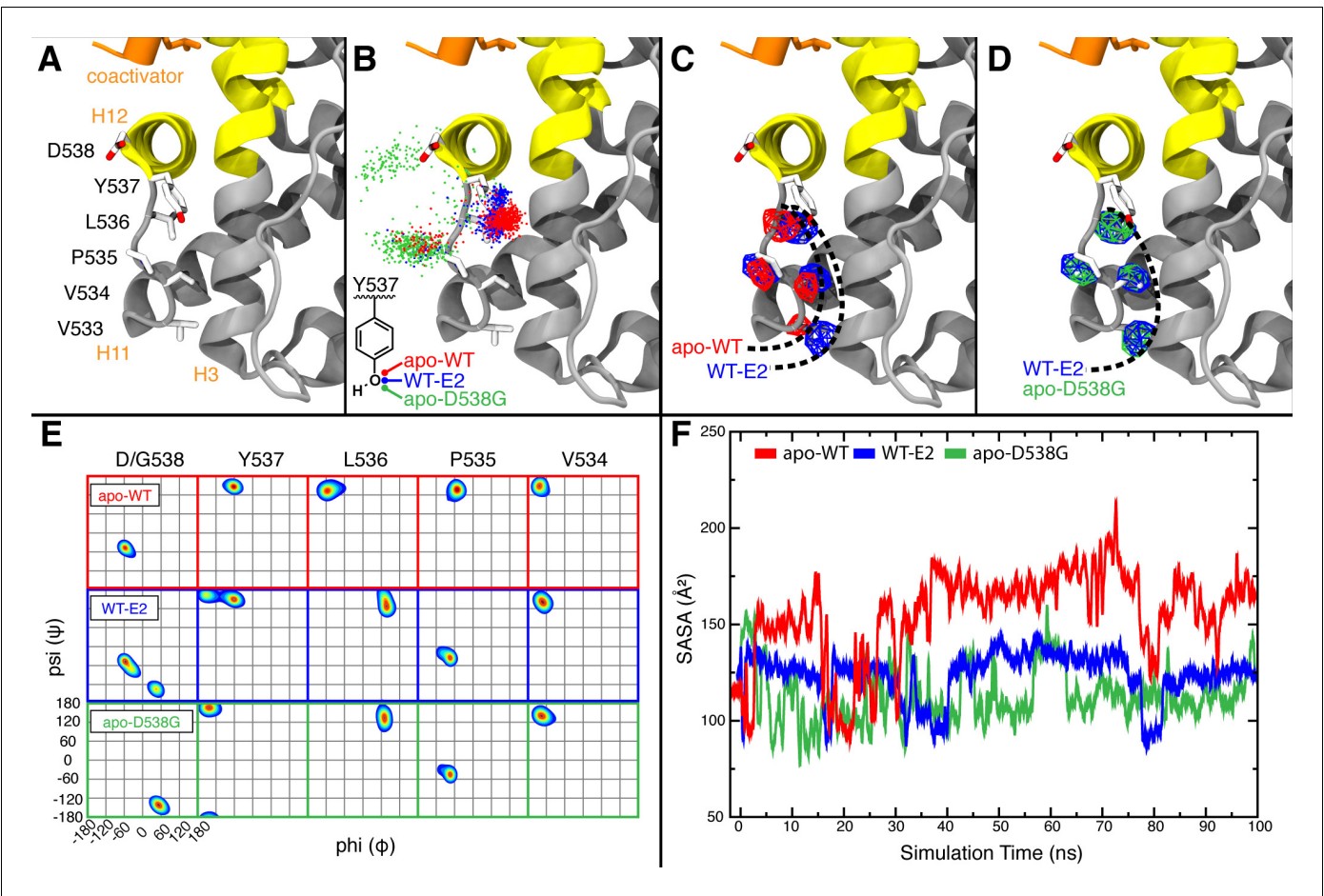

**Figure 6.** Visualization of H11-12 loop dynamics. (**A**) H11-12 loop of WT ERα LBD-E2 complex. (**B**) Superimposing the position of the phenolic oxygen of Y537 at 0.1-ns intervals for *apo* WT (red), WT-E2 (blue), and *apo* D538G mutant (green). (**C**) Mapping the mass density isosurface (0.75, i.e., 25th percentile) of the hydrophobic side chains in the linker region (V533, V534, P535, and L536). (**D**) Side-chain packing of the *apo* D538G structure compared to WT-E2. (**E**) Ramachandran analysis of residues 534–538 for the *apo* WT, WT-E2, and *apo* D538G MD simulations. (**F**) Time series of the solvent accessible surface area (SASA) for hydrophobic loop residues (533–536). LBD, ligand binding domain.
DOI: https://doi.org/10.7554/eLife.12792.027

MD simulations of the WT and the D538G mutant showed an increased flexibility of the H11-12 loop as a result of the D538G mutation, inducing the Y537 side chain to rotate toward the bulk solvent (*Figure 6B*). This rotation shifts the backbone conformations of residues 535–537 (*Figure 6E*) to occupy regions of the Ramachandran plot that are similar to WT-E2 and distinct from *apo*-WT. These mutation-induced changes allow the H11-12 loop to adopt conformations similar to WT-E2, despite the absence of ligand. Computing the density maps for the side-chain atoms of hydrophobic residues V533, V534, P535, and L536 further confirmed this altered state in which the resulting backbone conformation also permits new side-chain positions (*Figure 6C,D*). Analysis of the averaged atomic density for residues 533–536 in the WT simulations reveals that the removal of the ligand (WT-E2 vs. *apo* WT, *Figure 6C*) results in more exposed positions for the hydrophobic residues in the loop region, thus destabilizing the H11-12 loop, while the D538G mutation allows the receptor to maintain side-chain positions buried more deeply into the protein surface (WT-E2 vs. *apo* D538G, *Figure 6D*). Further, reduced fluctuations were exhibited in the WT-E2 and *apo*-D538G MD simulations, as observed from larger volumes for the given isosurface, thus indicating that the residues pack more favorably. The optimized packing of the hydrophobic loop residues was additionally quantified by the decreased solvent exposure for the WT-E2 and *apo* D538G conformations compared to *apo* WT over the course of the entire simulation (*Figure 6F*). All of the changes that result from replacing D538 with glycine are consistent with increased stability of the H11-12 loop in the mutant, which likely contributes to its constitutive activity.

## Structural and biophysical Basis for reduced SERM potency

### Trypsin susceptibility of the H12 mutants with TOT

Trypsin susceptibility was used to determine whether the antagonist state dynamics of the H11-12 loop and H12 were altered as a result of Y537S or D538G mutation. Interestingly, these regions showed decreased dynamics (i.e. increased stability) for the Y537S and D538G mutants, which displayed $t_{1/2}$ = 60 and 62 min, respectively, whereas the $t_{1/2}$ for the WT was 18 min (*Figure 4A*). These half-lives were higher than *apo* proteins alone suggesting that TOT binds and increases the overall stability of the protein (*Figure 4A*), although to a lesser extent than does E2.

### HDX MS of the WT and mutants in complex with TOT

HDX MS was employed to probe the sequence-specific conformational mobility of the Y537S and D538G antagonist states compared to the WT. Comparison of HDX profiles for TOT-bound WT and mutants revealed that the mutant proteins adopt alternate conformations in H11/12 regions relative to the WT complex (*Figure 7C–E*). *Figure 7—figure supplements 1–3* show deuterium uptake plots for the WT and mutant ERα LBDs in complex with TOT for the full protein sequence. Additionally, *Figure 7—figure supplements 4–6* show side-by-side comparisons for the WT, Y537S, and D538G ERα LBD in complex with ligand and/or SRC3 NRD versus their individual *apo* states.

### Structure of the D538G-TOT complex

To explore the structural basis for reduced SERM potency and efficacy, the D538G mutant ERα LBD was co-crystallized with TOT. We were unable to obtain diffraction-quality crystals for Y537S in complex with any SERM. However, the D538G-TOT structure was solved to 3.06 Å with four dimers in the ASU by molecular replacement. The TOT ligand and H12 are both well resolved in every monomer (*Figure 5—figure supplement 1B*). Significant conformational differences are observed between WT-TOT (PDB: 3ERT) and D538G-TOT structures, both in H12 and the H11-12 loop regions. We believe that these differences help account for the reduced potency and efficacy of TOT toward the D538G mutant ERα in breast cancer reporter gene assays.

As with the WT-TOT structure, H12 of the D538G-TOT structure lies in the AF-2 cleft; the conformation of H12 in the mutant structure, however, is altered compared to the WT (*Figure 7A*). In D538G-TOT, L536 is oriented toward solvent rather than docking into the well-defined leucine-binding pocket found in the WT-TOT structure, and P535 occupies the space previously occupied by the L536 of the WT (*Figure 7A*). The largest conformational change occurs in the H11-12 loop (residues 527–537). Instead of extruding toward solvent, the loop is packed toward the interior of the protein by 9.6 Å compared to the WT (V534 alphacarbon to alpha carbon) (*Figure 7A*). This conformational change likely explains why trypsin displayed a reduced ability to cleave at this region. Additionally,

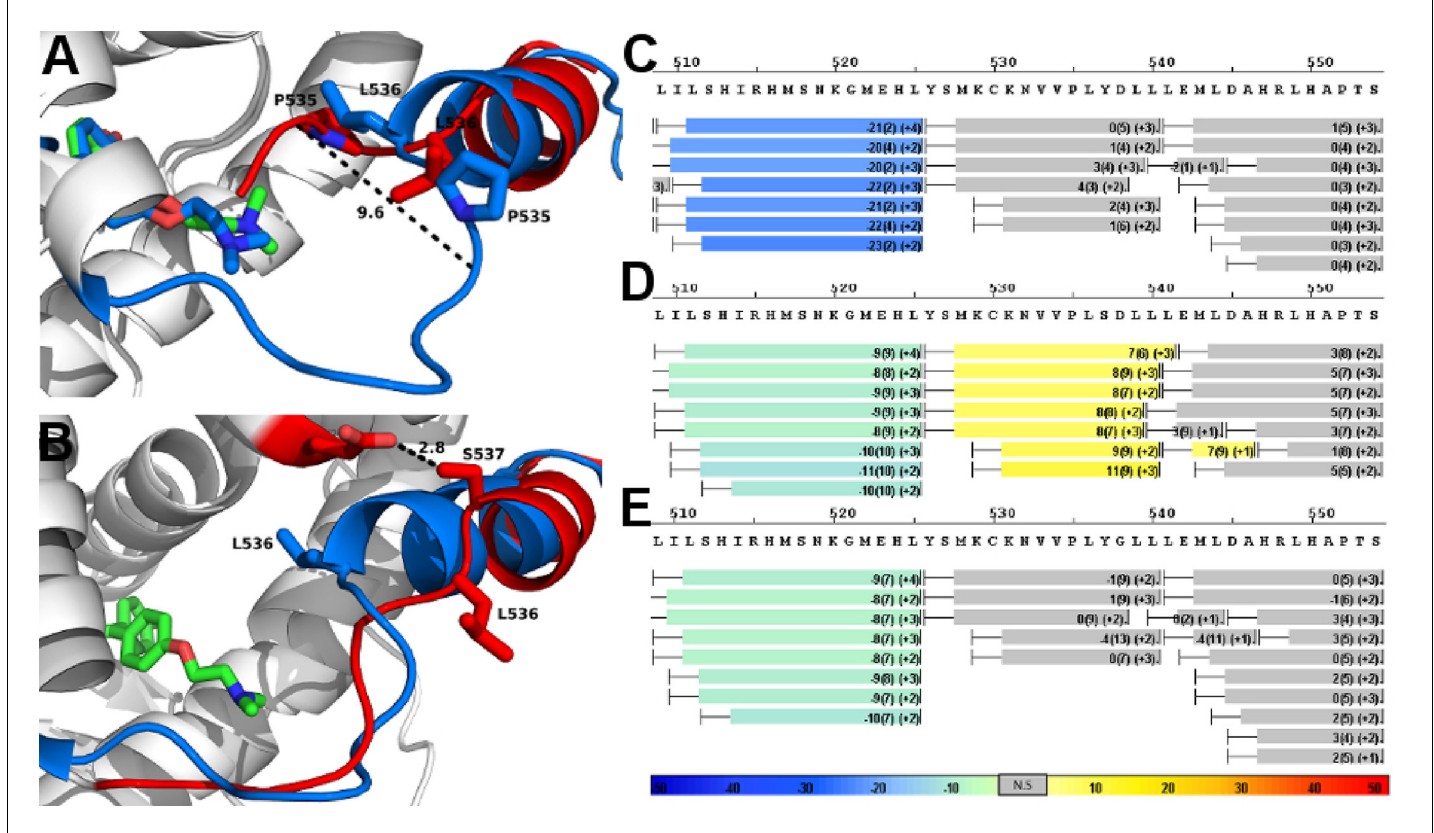

**Figure 7.** Alterations to the D538G and Y537S antagonist conformational states. (**A**) Superposition of monomer A for the 538G-TOT structure with the WT (3ERT). TOT and residues 530–550 of the WT (blue) (PDB: 3ERT), TOT of D538G (green), residues 531–550 (red). (**B**) Predicted conformational alterations in H12 in the Y537S-TOT structure (red) compared to the WT-TOT (blue). (**C**) HDX-MS of the WT-TOT complex for H11 through H12 regions. (**D**) HDX-MS of Y537S-TOT complex for H11 through H12 regions. (**E**) HDX-MS of the D538G-TOT complex for H11 through H12 regions. HDX data is color coded as in 2C. See methods for more details on coloring scheme. HDX-MS, hydrogen/deuterium exchange mass spectrometry; LBD, ligand binding domain.

DOI: https://doi.org/10.7554/eLife.12792.028

The following figure supplements are available for figure 7:

**Figure supplement 1.** Complete differential amide HDX-MS map of WT ERα LBD binding to TOT.

DOI: https://doi.org/10.7554/eLife.12792.029

**Figure supplement 2.** Complete differential amide HDX-MS map of Y537S ERα LBD mutant binding to TOT.

DOI: https://doi.org/10.7554/eLife.12792.030

**Figure supplement 3.** Complete differential amide HDX-MS map of D538G ERα LBD mutant binding to TOT.

DOI: https://doi.org/10.7554/eLife.12792.031

**Figure supplement 4.** Experiment comparison view comparing the differential HDX behavior of *apo* WT ERα LBD in the presence of various ligands or coactivator.

DOI: https://doi.org/10.7554/eLife.12792.032

**Figure supplement 5.** Experiment comparison view comparing the differential HDX behavior of *apo* Y537S ERα LBD in the presence of various ligands or coactivator.

DOI: https://doi.org/10.7554/eLife.12792.033

**Figure supplement 6.** Experiment comparison view comparing the differential HDX behavior of *apo* D538G ERα LBD in the presence of various ligands or coactivator.

DOI: https://doi.org/10.7554/eLife.12792.034

the tertiary amine at the terminus of the TOT ligand is observed in several conformations in the complex with D538G ER rather than the single conformation present in the WT-TOT structure. Together, these observations suggest that the flexibility of a glycine at position 538 reduces the ability of an antagonist to influence the H11-12 loop and H12. However, care must be taken when interpreting differences within this loop between the WT and the D538G mutant crystal structures. A crystal

contact is formed in the WT-TOT structure between the backbone amide of K531 with the backbone carbonyl oxygen of K492 in a symmetry mate. Together, these data reveal that the D538G mutant adopts an altered antagonist conformation that resists antagonism relative to the WT-TOT complex.

## Modeled structure of the Y537S-TOT complex

MD simulation was used to model Y537S with TOT because we were unable to obtain diffraction quality crystals for the complex. During the simulation, H12 of Y537S was found to lie within the AF-2 cleft in a perturbed conformation compared to the WT-TOT crystal structure, similar to that seen in the D538G-TOT crystal structure. Specifically, L536 no longer packs well with the leucine binding site on H3 but reorients to face the solvent, and the rest of the motif is also pushed outward and even shifted toward the C-terminus along the axial direction of H12 by half a turn (*Figure 7B*). These findings suggest that Y537S stabilizes H12 inside the AF-2 through the formation of a newly formed hydrogen bond (*Figure 7B*) that is predicted to form between S537 and E380. Like the D538G-TOT complex, our data for the Y537S-TOT complex show that these conformational changes serve to reduce the inhibitory potency of the SERM relative to the WT ERα LBD.

## Discussion

Acquired resistance to endocrine therapies represents a substantial barrier toward obtaining pro-longed remission of ER-dependent metastatic breast cancers for a significant population of patients. While somatic mutations in the androgen receptor are a known mechanism of acquired hormone therapy resistance in prostate cancer, somatic mutations in *ESR1* have only recently been identified as an important mechanism of acquired endocrine therapy resistance in breast cancer. Subsequent studies have established Y537S and D538G as the two most common point mutations conferring hormone-independent activation and reduced SERM/SERD/AI inhibitory potency and likely efficacy (*Robinson et al., 2013*; *Toy et al., 2013*; *Jeselsohn et al., 2014*). The clinical importance of these *ESR1* mutations highlights the importance of understanding the mechanisms by which they influence ERα structure and function.

Here, biochemical and biophysical techniques combined with x–ray crystal structures, and MD simulations provide a molecular explanation for how the Y537S and D538G point mutations in the ERα LBD alter the structure and function of the receptor. Coactivator binding assays show that these mutant LBDs recruit the SRC3 coactivator in the absence of hormone, while the unliganded WT LBD does not. Importantly, *apo* Y537S binds SRC3 NRD with a significantly increased affinity compared to D538G. This differential coactivator binding affinity likely accounts for the significantly increased constitutive transcriptional activity of Y537S versus D538G in breast cancer cell line reporter gene assays (*Toy et al., 2013*). *Figure 8* shows a model for aberrant ERα activity as a result of Y537S and D538G mutations in the recurrent anti-estrogen-resistant breast cancer cell. Ligand-binding assays demonstrate that both mutants possess a slightly reduced affinity for E2 and a significantly reduced affinity for TOT. Collectively, these data suggest that the combination of a recruitment of coactivator in the absence of hormone and a reduced TOT-binding affinity underlie the hormone therapy resistance conferred by these H12 ERα mutations.

Comprehensive biophysical and structural investigations by proteolytic susceptibility assays, HDX-MS, x-ray crystallography, and MD simulations reveal how the Y537S and D538G mutations affect ERα in the *apo*, agonist, and antagonist-bound states, thereby providing a detailed structural explanation for the hormone-resistance conferred to the ERα. The Y537S and D538G mutations are located at or near H12, a key molecular switch governing the ligand-regulated actions of ERα via AF-2. Previously published *apo* and agonist-bound Y537S structures showed that S537 promotes the agonist conformation in the absence of ligand by forming a hydrogen bond to D351 (*Nettles et al., 2008*), in the process facilitating a tighter packing of the H11-12 loop against the LBD. Similarly, our analysis of the agonist-bound and *apo* D538G structures show that this mutation relaxes the helical character at the start of H12, thereby also relaxing the H11-12 loop and improving the packing of its hydrophobic side chains. Importantly, our data also show that binding of coregulator (SRC3) further stabilizes H12 in the agonist conformation. While the Y537S and D538G mutants may work through different mechanisms, both stabilize the agonist state in the absence of hormone. The D538G mutation, however, appears to be less stabilizing, as reflected by the lower constitutive activity of D538G ERα in both biochemical and cell-based assays (*Toy et al., 2013*).

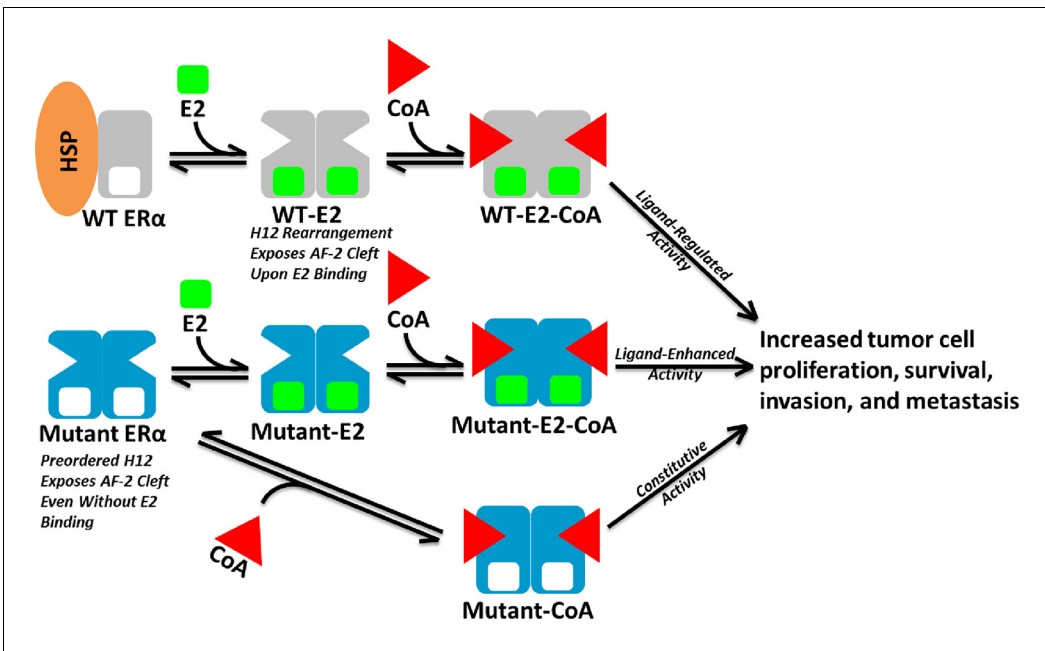

**Figure 8.** Model of Aberrent ERα Mutant Activity. Upon hormone binding (E2), WT ERα sheds heat-shock/ chaperone proteins (HSP), forms head-to-head homodimers, and recruits coactivator (CoA) to become active. By contrast, Y537S or D538G ERα mutants adopt the active conformation in the absence of hormone to recruit CoA and achieve constitutive activity. Additionally, E2 binding may further increase mutant activity.
DOI: https://doi.org/10.7554/eLife.12792.035

Examination of the molecular basis for reduced SERM potency and efficacy for mutant ERα LBDs reveals that this likely evolves from structural changes to the H11-12 loop, resulting in a decreased binding affinity of antagonist ligands and an altered, stabilized, antagonist conformation of H12 in the AF-2 cleft. Our biophysical studies indicate that the H11-12 loop and H12 are both altered when TOT is bound in the Y537S and D538G mutants compared to the WT. Further, when compared to the WT-TOT structure, the D538G-TOT structure shows an altered conformation of the H11-12 loop and H12 occupancy of the AF-2 cleft, and multiple conformations of the TOT ligand (indicative of reduced influence on the H11-12 loop). Additionally, MD simulation of the Y537S-TOT complex shows that S537 might form a hydrogen bond with E380 that alters the antagonist conformation. Therefore, the reduced inhibitory potency of TOT stems from its reduced affinity for the Y537S and D538G mutants along with conformational changes to the antagonist state once it occupies the ligand-binding site.

Taken together, these results suggest that the constitutive activity conferred by the Y537S and D538G mutations stems from the intrinsic ability of the mutant receptors to adopt a stable agonist conformation in the absence of hormone, thereby leading to enhanced recruitment of SRC3 coactivators and increased ERα transcriptional activity. This pre-organized agonist state contributes to a decreased affinity for hormone and especially for SERMs because the stabilized H12 agonist conformation restricts ligand access to the hormone-binding pocket. In addition to reduced ligand affinity, SERM action is further reduced by an altered antagonist state of H12. Thus, recruitment of coactivators in the breast cancer cell is not inhibited as efficiently for the Y537S and D538G mutants as for WT ERα.

One caveat to the approach described in this study is that ERα is a multi-domain protein and only the LBD was used for structural studies. To gain deeper insight into how these mutations affect full length ERα, further studies on intact multi-domain protein will be necessary. In addition, the effect of these mutations on the other aspects of ERα action including other hormone/SERM/SERD binding affinities, homodimer formation, DNA-binding, and stability (in vitro and in vivo) and whether these mutant receptors display a differential preference for a spectrum of coactivators must be investigated.

Our findings suggest that SERMs and SERDs that are designed to specifically increase the dynamics of H12 might lead to drugs with increased potency. In this regard, our data show that the H11-12 loop plays an important and previously unrecognized role in regulating the behavior of H12, an essential molecular switch that is allosterically controlled by ligand, which determines the differential ability of the ERα AF-2 to recruit coactivators and corepressors. Therefore, antagonists with improved inhibitory potency will increase the dynamic character of mutant H12, an already appreciated aspect of SERD action (*Pike et al., 2001*). Additionally, our work provides a biophysical hypothesis for why fulvestrant (a SERD, known to disorder H12) was the only molecule which could completely ablate the transcriptional activity of the Y537S and D538G mutants in breast cancer cells while TOT (a SERM) could not (*Toy et al., 2013*). Therefore, newly developed mixed SERM/SERDs and SERDs with improved pharmacokinetics and oral bioavailability over fulvestrant, such as AZ9496, bazedoxifene, GDC910, and RAD1901, should be particularly effective against cancers expressing the Y537S and D538G *ESR1* mutants (*De Savi et al., 2015*; *Garner et al., 2015*; *Lai et al., 2015*; *Wardell et al., 2013*). These compounds may prove invaluable for treating endocrine therapy-resistant ER+ breast cancers and also preventing or delaying the appearance of these somatic mutations in early-stage patients.

## Materials and methods

### Time resolved-FRET assays

#### Protein preparation for TR-FRET

Site-directed mutagenesis was used to generate the Y537S and D538G mutations in the LBD of the human estrogen receptor α (ERα amino acids 304–554). The WT and mutant ERα and the nuclear receptor domain (NRD) of human SRC3 encompassing three NR boxes (amino acids 627–829) were expressed in *E. coli*, using methods reported previously (*Jeyakumar et al., 2011*; *Carlson et al., 1997*). ER LBDs of wild type, Y537S, and D538G were prepared as 6×His fusion proteins, with a single reactive cysteine at C417. While bound to the Ni-NTA-agarose resin (Qiagen Inc., Santa Clarita, CA), the ERs were labeled with MAL-dPEG4-biotin (Quanta BioDesign, Powell, OH), site-specifically at C417. The SRC3-NRD construct has 4 cysteines and was labeled non-specifically, also while on the resin, with 5-iodoacetamido fluorescein (Molecular Probes, Invitrogen, Eugene, OR). It was previously determined that an average of 1.8–2 fluorescein molecules are attached to the SRC3 NRD (*Kim et al., 2005*).

#### SRC titration

SRC3 was titrated into a fixed amount of ERα-LBD-biotin mixed with SaTb (streptavidin-terbium, Invitrogen, Grand Island, NY), on 96-well black microplates (Molecular Devices, Sunnyvale, CA) following previously determined methods (*Jeyakumar et al., 2011*). The time-resolved Förster resonance energy transfer (tr-FRET) measurements were performed with a Victor X5 plate reader (Perkin Elmer, Shelton, CT) with an excitation filter at 340/10 nm and emission filters for terbium and fluorescein at 495/20 and 520/25 nm, respectively, with a 100 μs delay. Diffusion-enhanced FRET was determined by a parallel incubation without biotinylated ER-LBD and subtracted as a background signal. The final concentrations of reagents were: 1 nM ERα-417, 0.25 nM streptavidin-terbium, 1 μM ligand, SRC3-F1 coactivator titrated from $3.2 \times 10^{-7}$ to $3.2 \times 10^{-12}$M. The data, representing 2–3 replicate experiments, each with duplicate points, were analyzed using GraphPad Prism 4 and are expressed as the $EC_{50}$ in nM.

#### Ligand titration

Ligands were titrated into a constant amount of ER-LBD-biotin, SaTb, SRC3-F1. The final concentrations were 1 nM ER-LBD, 0.25 nM SaTb, 100 nM SRC3-fluorescein, and increasing ligand concentrations from $1 \times 10^{-12}$ to $1 \times 10^{-6}$ M. Diffusion-enhanced FRET was determined by a parallel incubation without biotinylated ER-LBD and subtracted as a background signal. The tr-FRET was measured with a Victor X5 plate reader as outlined above. The data, representing 2–3 replicate experiments, each with duplicate points, was analyzed using GraphPad Prism 4, and are expressed as the $EC_{50}$ in nM.

## Ligand-binding assays

Relative binding affinities (RBA) were determined by a competitive radiometric binding assay with 2 nM [$^3$H]-E2 as tracer, as a modification of methods previously described (*Katzenellenbogen et al., 1973*: *Carlson et al., 1997*). Incubations were at 0°C for 18–24 hr. Hydroxyapatite was used to absorb the receptor-ligand complex, and unbound ligand was washed away. The determination of RBA values is reproducible in separate experiments with a CV of 0.3. The IC$_{50}$ values for inhibition of [$^3$H]-E2 were converted to K$_i$ values using the Cheng-Prusoff equation (K$_i$ = IC$_{50}$/(1 + conc. tracer/K$_d$ tracer))(*Cheng and Prusoff, 1973*); this was necessary because the affinity of the [$^3$H]-E2 tracer is different for WT and mutant ERs. The K$_d$ of [$^3$H]-E2 for the ERs was determined in a saturation-binding assay, as 0.26 ± 0.13 nM for the WT, 1.43 ± 0.55 nM for Y537S, and 1.30 ± 0.63 nM for D538G (*Figure 2*). For the saturation ligand binding (Scatchard analysis), protein was diluted to 0.8 nM, in Tris-glycerol buffer (50 mM Tris pH 8.0, 10% glycerol, with 0.01 M 2-mercaptoethanol and 0.3 mg/mL ovalbumin added) and incubated with various concentrations of [$^3$H]-E2 (Perkin-Elmer, Waltham, MA) in the absence or presence of a 100-fold excess of unlabeled ligand for 3–4 hr, at 0°C. Aliquots of the incubation solution were used to determine the total [$^3$H]-E2 in the sample. The incubation solutions were then assayed by adsorption onto HAP (hydroxyapatite, BioRad, Hercules, CA) and the free estradiol was washed away. Data were processed by GraphPad Prism 4 according to the method of Scatchard (*Scatchard, 1949*; *Hurth et al., 2004*).

## Trypsin proteolysis

Protein was prepared and labeled as described above for the trFRET assays. It was incubated in t/g buffer with or without 1 µM of ligand, at room temperature for 1 hr. Then, 1 µg trypsin per unit of protein was added for 10, 30, 60, and 300 min at room temperature according to previously established methods (*Tamrazi et al., 2005*). FRET signal was measured using a Victor X5 plate reader as outlined above. The data, representing 2–3 replicate experiments, were analyzed using GraphPad Prism 4, and are expressed as half-lives (t$_{1/2}$).

## Hydrogen deuterium exchange

### Differential hydrogen/deuterium exchange (HDX) MS

Solution-phase amide HDX experiments were carried out using a fully automated system as described previously with slight modifications.(*Chalmers et al., 2006*) Prior to HDX, 10 µM of 6×-HIS-ERα-LBD (WT or mutants) were incubated with 100 µM of individual ligands for 1 hr on ice for complex formation. Differential HDX experiments with ligands were initiated by mixing either 5 µl of the ERα LBD alone (*apo*) or the complex (1:10 molar mixture of ERα and ligands) with 20 µl of D$_2$O-containing HDX buffer (20 mM Tris 8.0, 150 mM NaCl, and 3 mM DTT). For the differential HDX experiments with SRC3 NRD, 10 µM of WT or mutant ERα LBDs were mixed with 25 µM of SRC3 NRD for 2 hr on ice for complex formation and then subjected to HDX as described above. For the *apo* ERα comparisons, 10 µM of WT or mutant ERα LBDs were run in a similar differential format comparing either Y537S or D538G directly with the WT. Twenty-five microliter aliquots were drawn after 0 s, 10 s, 30 s, 60 s, 900 s or 3,600 s of on-exchange at 4°C and the protein was denatured by the addition of 25 µl of a quench solution (1% v/v TFA in 5 M urea and 50 mM TCEP). Samples were then passed through an immobilized pepsin column at 50 µl min$^{-1}$ (0.1% v/v TFA, 15°C). and the resulting peptides were trapped on a C$_8$ trap column (Hypersil Gold, ThermoFisher, Grand Island, NY). The bound peptides were then gradient-eluted (5–50% CH$_3$CN w/v and 0.3% w/v formic acid) across a 1 mm × 50 mm C$_{18}$ HPLC column (Hypersil Gold, ThermoFisher, Grand Island, NY) for 8 min at 4°C. The eluted peptides were then subjected to electrospray ionization directly coupled to a high-resolution Orbitrap mass spectrometer (LTQ Orbitrap XL with ETD, Thermo Fisher).

### Peptide identification and HDX data processing

MS/MS experiments were performed with a LTQ linear ion trap mass spectrometer (LTQ Orbitrap XL with ETD, Thermo Fisher) over a 70-min gradient. Product ion spectra were acquired in a data-dependent mode and the five most abundant ions were selected for the product ion analysis. The MS/MS *.raw data files were converted to *.mgf files and then submitted to Mascot (Matrix Science, London, UK) for peptide identification. Peptides included in the peptide set used for HDX detection had a MASCOT score of 20 or greater. The MS/MS MASCOT search was also performed against a

decoy (reverse) sequence, and false positives were ruled out. The MS/MS spectra of all the peptide ions from the MASCOT search were further manually inspected, and only the unique charged ions with the highest MASCOT score were used in estimating the sequence coverage. The intensity-weighted average m/z value (centroid) of each peptide isotopic envelope was calculated with the latest version of our in-house software, HDX Workbench (*Pascal et al., 2012*). HDX data are presented as an average of three independent triplicates. Deuterium uptake for each peptide is calculated as the average of% $D_2O$ for the six time points (10 s, 30 s, 60 s, 300 s, 900 s, and 3600 s) and the difference in average% $D_2O$ values between the *apo* and liganded states is presented as a heat map with a color code given at the bottom of each figure (warm colors for deprotection and cool colors for protection) and colored only if they show a>5% difference (less or more protection) between the two states and if atleast two time points show a statistically significant difference in a paired two-tailed student's t-test (p<0.05). Grey color represents no significant change (0–5%) between the two states. The exchange at the first two residues for any given peptide is rapid and is ignored in the calculations. Each peptide bar in the heat map view displays the average Δ% $D_2O$ values with its associated standard deviation and the charge state shown in parentheses.

## X-ray crystallographic analysis of the D538G ERα LBD

### Generation and production of the D538G ERα LBD mutant

Quick Change Mutagenesis (New England Biolabs, Ipswitch, MA) was performed to change aspartate 538 to glycine on a pGM6 containing the gene for the 6×His-Tobacco etch virus (TEV)-ERα LBD. The following oligonucleotide primers were used to generate the mutant:

Forward: (*5'GGTGCCCCTCTACGGCCTGCTGCTGG3'*)
Reverse: (*5'CCAGCAGCAGGCCGTAGAGGGGCACC3'*)
The sequence for the resulting ERα LBD D538G mutant was verified by DNA sequencing.

### Protein expression for crystal generation

A 250 mL LB broth containing 100 µg/mL ampicillin was inoculated with a single colony of the *E. coli* expression strain BL21 (DE3) transformed with pGM6-ERα LBD D538G mutant. Following overnight incubation at 37°C, 10×1L LB broth containing 100 µg/mL ampicillin were each inoculated with 5-mL aliquots of the overnight culture. Cells grew at 37°C with shaking at 180 rpm until they reached mid-log phase growth ($OD_{600}$ = 0.8) at which point expression of the protein was induced with 0.3 mM IPTG and incubation continued overnight with shaking at 20°C. Cells were harvested by centrifugation at 3500 *g* for 30 min, and the pellet was frozen at -20°C. The pellet was resuspended in 200 mL BPER and 100 µg DNAse, protein inhibitor cocktail, and lysozyme were added to the lysate. Following 30 min of stirring at 4°C, the lysed cells were centrifuged at 22,000 *g* for 30 min and the supernatant isolated. The soluble fraction was incubated with 2 mL of pre-washed Ni-NTA resin (ThermoFisher, Grand Island, NY) then placed onto a column. The column was washed with 10 column volumes of buffer containing 20 mM Tris pH 8.0, 500 mM NaCl, 40 mM imidazole pH 8.0, 10% glycerol, and 15 mM 2-mercaptoethanol, and the protein was subsequently eluted from the column using a buffer containing 20 mM Tris pH 8.0, 500 mM NaCl, 500 mM imidazole pH 8.0, 10% glycerol, and 15 mM 2-mercaptoethanol. The 6×His-TEV tag was removed using a 15:1 w/w ratio of LBD to TEV protease. The LBD was isolated from the tag by a pass over a column containing 2 mL of washed Ni-NTA resin and the flow through, containing the LBD, was isolated. The protein was dialyzed overnight in a buffer containing 20 mM Tris pH 8.0, 20 mM NaCl, 10% glycerol, and 15 mM 2-mercaptoethanol then subjected to a final purification on a Resource Q ion exchange column (ThermoFisher, Grand Island, NY). A 100 mL linear gradient was used to elute the protein with a buffer containing 20 mM Tris pH 8.0, 500 mM NaCl, 10% glycerol, and 15 mM 2-mercaptoethanol. A single peak corresponding to the ERα LBD D538G mutant was isolated and a single band was observed on a SDS-PAGE gel (BioRad, Hercules, CA). Lastly, the LBD was concentrated to 10 mg/mL using a spin concentrator, separated into 100-µL aliquots, flash frozen, and stored at -80°C until use.

### Crystallization of the ERα LBD D538G mutant

For the estradiol (E2) and 4-hydroxytamoxifen (TOT)-bound structures, the purified ERα LBD D538G mutant at 10 mg/mL was incubated for overnight with 1 mM ligand. For the *apo* D538G and E2 structures, a 2.5-fold mol:mol (excess) of glucocorticoid receptor interacting protein NR box II

peptide (GRIP) was incubated with the LBD for approximately 3 hr. Hanging drop method was used for all crystals using VDX pre-greased plates (Hampton Research, Aliso Viejo, CA). For the *apo* D538G structure, 15 mM MgCl$_2$ and 10 mM ATP were added to the protein prior to plating. A total of 1 μL of 5 mg/mL *apo* D538G was mixed with 1 μL of 30% PEG 3350, 200 mM MgCl$_2$, and 100 mM Tris pH 8.5. For the E2-complex structure a total of 1 μL of 5 mg/mL protein was mixed with 1 μL of 25% PEG 3,350, 200 mM MgCl$_2$, 100 mM Tris pH 8.5, and 1 mM phenylalanine. For the D538G-TOT complex structure, the protein/ligand was centrifuged at 19000 *g* to remove precipitate then 2 μL at 10 mg/mL was mixed with 2 μL of 400 mM ammonium sulfate, 100 mM Tris pH 8.0, and 10% glycerol. For the *apo* and E2-bound structures, clear triangular rods appeared after 3 days. For the TOT-bound structure, clear rectangular rods appeared overnight. Paratone-N was used as the cryo-protectant for the *apo* and TOT-bound structures, whereas 25% glycerol was used as the cryo-protectant for the E2-bound structure. All x-ray data sets were collected at the Advanced Photon Source at Argonne National Laboratories, Argonne, Illinois. The TOT-complex data set was collected at the SBC 19-BM beamline (0.97 Å), the E2-bound structure at LS-CAT 21-ID-D (0.97 Å), and the *apo* structure at LS-CAT 21-ID-F (0.97 Å).

### X-ray structure solution

Data were indexed, scaled and merged using HKL-3000(*Otwinowski and Minor, 1997*). Phaser was used for all molecular replacements (*McCoy et al., 2007*). An existing structure of the WT ERα LBD in complex with TOT (PDB: 3ERT) was modified by removing all ligands and water molecules, and then used as the search molecule for the D538G-TOT structure (*Shiau et al., 1998*). For the WT and *apo* D538G structures, an existing WT ERα LBD-agonist structure (PDB: 2QXM) was modified by removing all ligands and water molecules, and then used as the search molecule (*Nettles et al., 2008*). For the *apo* and E2-bound structures, one dimer was found in the asymmetric unit (ASU), whereas four dimers were found for the TOT-bound structure. The CCP4i (Refmac) program suite was used for all refinement (*Winn et al., 2011*). The models were refined using iterative rounds of Refmac and Coot. Densities for the ligands were clearly visible after the first round of refinement for both the E2- and TOT-bound structures. Unresolved residues were not included in the structures deposited in the Protein Data Bank including the *apo* D538G (PDB: 4Q13), D538G-E2 complex (PDB: 4PXM), and D538G-4OHT (PDB: 4Q50) structures. All x-ray crystal structure images were made using Pymol.

## Molecular dynamics simulations of D538G

### Structure preparation

Atomistic molecular models of dimeric ERα were constructed in silico starting from an x-ray crystal structure of ERα in complex with E2 and a coactivator peptide ( Atomic coordinates were downloaded from the Protein Data Bank (PDB code: 1GWR) (*Wärnmark et al., 2002*)and prepared using a combination of the MOE (*Molecular Operating Environment, 2014*) and VMD (Visual Molecular Dynamics; *Humphrey et al., 1996*). Using the Structure Preparation module within MOE, all missing loops were constructed, explicit hydrogen atoms added, a side-chain rotamer search was performed, and protonation states were computed for all titratable residues. The resulting structure was loaded into VMD, where each protein monomer, coactivator peptide, and all crystallographic water molecules were written to separate PDB files; the E2 ligand coordinates were discarded for simulated *apo* structures. Each histidine residue was renamed according to the CHARMM naming convention to reflect the computed protonation states, as shown in *Table 4*. The dimeric ERα structure was then constructed from the separate PDB files using the PSFGen plugin within VMD. The N- and C-termini were capped with neutral acetyl and *N*-methylamido groups, respectively. The protein complex was subsequently solvated using the Solvate plugin of VMD with a 20-Å padding thickness on all sides, and ions were added using the Autoionize plugin to neutralize the system and yield a final NaCl concentration of 0.1 M. Ions were placed a minimum distance of 5 Å from the protein surface. The resulting fully solvated system contained ~101k atoms. The D538G mutant structure was constructed in an analogous manner, differing only in an additional mutate' command in PSFGen to create the D538G mutation. Additional steps to minimize and equilibrate the mutated region are discussed below.

**Table 4.** Protonation states of histidines for the structure used in MD simulations.

| HIS residue number | Monomer A | Monomer B |
|---|---|---|
| 356 | HSE | HSD |
| 373 | HSD | HSE |
| 377 | HSE | HSD |
| 398 | HSP | HSP |
| 474 | HSE | HSE |
| 476 | HSE | HSE |
| 488 | HSE | HSE |
| 501 | HSD | HSE |
| 513 | HSD | HSD |
| 516 | HSE | HSE |
| 524 | HSE | HSE |
| 547 | HSE | HSE |

DOI: https://doi.org/10.7554/eLife.12792.036

## Simulations

All MD simulations were performed using the NAMD2 software package (*Phillips et al., 2005*). The CHARMM36 force field was used to describe the protein, solvent, and ions, and included CMAP backbone corrections and NBFIX terms for protein-ion interactions (*Mackerell et al., 1998*; *Mackerell, 2004*). The TIP3P water model was used to as the explicit solvent (*Jorgensen et al., 1983*). Ligand parameters for E2 were taken from the CHARMM General Force Field (CGenFF; *Vanommeslaeghe and MacKerell, 2012*) as assigned by analogy using the ParamChem (*Vanommeslaeghe and MacKerell, 2012*) webserver. Attempts to further refine torsion parameters with moderate penalty scores using the Force Field Toolkit (ffTk; *Mayne et al., 2013*) did not yield significant improvement of the potential energy surface. Simulations were performed under an NPT ensemble at 1.0 atm and 310 K, employing a Nosé-Hoover thermostat and a Langevin piston with a period of 100 fs, decay of 50 fs, and damping coefficient of 0.5 ps$^{-1}$ (*Martyna et al., 1994*; *Feller et al., 1995*). A simulation time step of 2 fs was used, and atomic coordinates were recorded every 500 steps (1 ps). The molecular system employed periodic boundary conditions, and non-bonded interactions were truncated using a switching function from 10.0 to 12.0 Å. Long-range electrostatics were evaluated using the particle mesh Ewald (PME) method (*Darden et al., 1993*). Bonded and non-bonded forces were computed at every time step, while PME forces were computed every other time step.

All molecular systems were first simulated to equilibrate 'non-natural' components of the system by applying harmonic restraints ($k$ = 1 kcal/mol/Å$^2$) on heavy atoms present in the 1GWR x-ray crystal structure. Atoms belonging to added water, ions, missing loops (± 2 residues), or mutated residues (± 2 residues) were left unrestrained. The system was subjected to a 10000-step downhill minimization, followed by 1 ns of simulation. All restraints were then released, and the system was simulated for an additional 100 ns of production simulation.

## MD simulation trajectory analysis

All analyses were performed using VMD (*Humphrey et al., 1996*). Simulation trajectories were first prepared by removing water molecules, concatenating sequential trajectory files, downsampling the framerate to 10 ps/frame, and rewrapping the periodic system to move the protein center of mass to the center of the periodic cell. Prior to analysis, all trajectories were aligned to the initial frame by fitting Cα atoms of the protein, excluding the coactivator peptides from the fit measurement. When a consistent reference frame was required for cross-trajectory comparisons, all frames were aligned to the 1GWR x-ray structure prior to analysis. With the exception of explicit time series measurements (i.e. SASA), all other analyses were performed for the last 50 ns of the 100-ns production simulation.

Side-chain conformations of residue Y537 were visualized by superimposing the position of the phenolic oxygen every 100 ps (n = 500) using the standard 'points' representation of VMD. Density maps of side chain and backbone atoms were computed using the Volmap plugin of VMD with a resolution of 1 Å and averaging the mass-weighted density over the trajectory. The volumetric maps for visualizing the side-chain positions were set to the 0.75 isosurface, representing the volume containing atomic density for greater than 75% of the analyzed trajectory. Ramachandran analysis was performed by measuring the $\varphi$ and $\psi$ dihedral angles for each residue at a 10-ps interval (n = 5000). The data were then converted to a two-dimensional histogram and plotted using the Matplotlib package of the python programming language (*Hunter, 2007*). A Gaussian filter was used to smooth the data ($\sigma = 10.0$), and the resulting bins were grouped into 10 contours. The lowest intensity contour (background, dark blue) was removed for clarity. The solvent accessible surface area (SASA) was computed for the side chains of hydrophobic residues 533–536 using the built-in measure sasa function of VMD. The default probe radius of 1.4 Å was used while taking the surrounding protein environment into account. SASA measurements were computed at 10-ps intervals (n = 10000) over the entire production simulation and smoothed using a Gaussian-weighed running average ($\sigma = 10.0$).

## Molecular dynamics simulations of Y537S-TOT complex

A parameter set was constructed for TOT. Its structure was optimized quantum mechanically at the level of restricted Hartree-Fock (RHF) 6-31g* using the computer program Gaussian 03 (Gaussian 03, Revision C.02, *Frisch et al., 2004*). The partial atomic charges of TOT were then derived with Restrained ElectroStatic Potential (RESP) (*Bayly et al., 1993*; *Cornell et al., 1993*) fitting to the quantum mechanical RHF/6-31g* potential. The ideal geometry was defined as the optimized. The other molecular mechanical parameters were derived by assigning CHARMm22 atom types for TOT (*Momany and Rone, 1992*).

 The dimer with the least missing residues of the H11-H12 loop was selected from the D538G-TOT crystal structure and served as the template structure to model the Y537S-TOT dimer structure. The side-chain atoms at positions 537 and 538 were removed, and then desired side-chain atoms were placed with the other missing atoms using the default geometry parameters in CHARMm22. Hydrogen atoms were placed with the hbuild module of the computer program CHARMM (*Brünger and Karplus, 1988*; *Vanommeslaeghe and MacKerell, 2012*). Missing residues (loops) in the starting crystal structure were optimized in three rounds (100 steps of the steepest descent method followed by two rounds of 100 steps of the adopted New-Raphson method) with updated harmonic constraints on the other atoms. Then, all newly added atoms' positions were optimized in the same fashion.

The resulting minimized structure was solvated with water molecules of 15 Å padding thickness from the molecular boundary and ionized to reach charge neutrality and the concentration of 0.145 M, both of which were done with VMD (*Humphrey et al., 1996*). The system was minimized for 5000 steps before a 100-ns MD simulation using NAMD2 (*Phillips et al., 2005*) was performed.

## Acknowledgements

The authors acknowledge funding from the Department of Defense Breast Cancer Research Program Breakthrough Award BC131458P1, the Susan G Komen Foundation PDF14301382 (to SWF), NIH P41-GM104601 (to ET), NIH T32ES007326 (to TAM), NIH R01-DK015556 (to JAK), NIH U54-MH084512 (to PRG PI: Rosen), Landenberger Foundation (to VD), and NSF CCF-1546278 (to YS). Supercomputing resources were provided through NSF XSEDE allocations (MCB104105 and MCB140135 to JAK). Viginia and DK Ludwig Fund for Cancer Research (to GLG). All x-ray crystal structure data sets were collected at Argonne National Laboratory, Structural Biology Center and the Life Sciences Collaborative Access team at the Advanced Photon Source. Argonne is operated by UChicago Argonne, LLC, for the US Department of Energy, Office of Biological and Environmental Research under contract DE-AC02-06CH11357. Special thanks to Dr. David Hosfield for critical reading and suggested edits.

## Additional information

### Funding

| Funder | Grant reference number | Author |
|---|---|---|
| Susan G. Komen for the Cure | Postdoctoral Fellowship | Sean W Fanning |
| U.S. Department of Defense | Breast Cancer Research Program Breakthrough Award | Sean W Fanning<br>Bradley Green<br>Srinivas Panchamukhi<br>Yang Shen<br>Sarat Chandarlapaty<br>Geoffrey L Greene |
| National Institutes of Health | R01 | Kathryn E Carlson<br>Teresa A Martin<br>John A Katzenellenbogen |
| National Institutes of Health | Postdoctoral Training Grant | Teresa A Martin |
| National Institutes of Health | P41 | Emad Tajkhorshid |
| National Science Foundation | CFF | Yang Shen |
| National Science Foundation | XSEDE | John A Katzenellenbogen |
| Virginia and D.K. Ludwig Fund for Cancer Research | | Geoffrey L Greene |

The funders had no role in study design, data collection and interpretation, or the decision to submit the work for publication.

### Author contributions

Sean W Fanning, Conducted and analyzed all x-ray crystallographic experiments, directed, coordinated, and interpreted all other experiments, conceived, wrote and submitted the manuscript, conception and design, acquisition of data, analysis and interpretation of data, drafting or revising the article, contributed unpublished essential data or reagents; Christopher G Mayne, Performed molecular dynamics simulations for the unliganded and apo states, assisted in analysis and interpretation of the data, also helped to write and revise the paper, acquisition of data, analysis and interpretation of data, drafting or revising the article; Venkatasubramanian Dharmarajan, Performed all HDX MS experiments, assisted in data interpretation and revising the article, acquisition of data, Analysis and interpretation of data, drafting or revising the article; Kathryn E Carlson, Performed ligand binding, trypsin susceptibility, and coregulator binding experiments, acquisition of data, analysis and interpretation of data, drafting or revising the article; Teresa A Martin, Performed ligand binding, trypsin susceptibility, and coregulator binding experiments; Scott J Novick, Assisted in HDX experiments, acquisition of data; Weiyi Toy, Provided expertise and guidance throughout the experiments, conception and design, analysis and interpretation of data; Bradley Green, Srinivas Panchamukhi, Provided reagents for ligand binding experiments, contributed unpublished essential data or reagents; Benita S Katzenellenbogen, Contributed knowledge/expertise and edits to the manuscript, conception and design, Analysis and interpretation of data, drafting or revising the article; Emad Tajkhorshid, Provided computational expertise for all MD simulations, also provided guidance and revisions to the manuscript, conception and design, analysis and interpretation of data, drafting or revising the article; Patrick R Griffin, Provided guidance and expertise for all HDX MS experiments, also provided guidance and revisions for the manuscript, analysis and interpretation of data, drafting or revising the article; Yang Shen, Performed MD simulations for TOT binding to the Y537S mutant, also provided critical reading and edits to the manuscript, acquisition of data, analysis and interpretation of data, drafting or revising the article; Sarat Chandarlapaty, Provided guidance throughout the experimental design and execution process, also provided critical guidance and edits to the manuscript, conception and design, analysis and interpretation of data, drafting or revising the article; John A Katzenellenbogen, Oversaw ligand-binding, coregulator-binding, and trypsin protealysis experiments, provided critical reading and revisions to the manuscript, conception and design, analysis and interpretation of data, drafting or revising the article; Geoffrey L Greene, Oversaw and guided the structural aspects of the work, helped Dr. Fanning coordinate all other experiments and

facilitated collaborations, drovided critical reading and revisions to the manuscript, conception and design, nalysis and interpretation of data, drafting or revising the article

## Author ORCIDs
Sean W Fanning http://orcid.org/0000-0002-9428-0060
Christopher G Mayne http://orcid.org/0000-0001-8905-6569
Bradley Green http://orcid.org/0000-0003-4106-285X
Emad Tajkhorshid http://orcid.org/0000-0001-8434-1010
Geoffrey L Greene http://orcid.org/0000-0001-6894-8728

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
