## [Decision Letter]

Thank you for submitting your work entitled "ESR1 Mutations Y537S and D538G Confer Breast Cancer Endocrine Resistance by Stabilizing the AF-2 Binding Conformation" for consideration by *eLife*. Your article has been reviewed by three peer reviewers, one of whom is a member of our Board of Reviewing Editors, and the evaluation has been overseen by Peter Tontonoz (Reviewing Editor) and Charles Sawyers as the Senior Editor. The other two reviewers, Donald McDonnell and Myles Brown, have also agreed to share their names.

The reviewers have discussed the reviews with one another and the Reviewing Editor has drafted this decision to help you prepare a revised submission.

The reviewers agree that this is a compelling study addressing the molecular mechanisms underlying the altered transcriptional activity of two ERα mutants commonly found in advanced metastatic breast cancer. The authors, evaluate the hypothesis that mutations in the H11/12 region of ERα alter the interaction of the receptor with transcriptional coregulators and that the basal, ligand independent, transcriptional activity of these mutants can likely be ascribed to their increased affinity for LXXLL-containing coregulators. The studies also explain the reduced affinity of these mutants for estradiol and tamoxifen. In all this is a tour de force structural analysis that addresses an important emerging issue in breast cancer.

Only minor concerns with the work were raised:

1) Most importantly, the reviewers felt that it would be desirable for the paper to be as accessible as possible for the general readership of *eLife*. The presentation is extremely dense in the description of the results and the biochemical figures which are hard to read because of the multiple curves.

2) In addition, a model figure would make the results more accessible to the general reader.

3) Finally, the authors should discuss the implications of their work for future drug discovery and comment on how their information could predict the likely efficacy of the most contemporary SERMs/SERDs i.e. GDC910, RAD1901, AZ9694.

---

## [Author Response]

*[…] In all this is a tour de force structural analysis that addresses an important emerging issue in breast cancer. Only minor concerns with the work were raised.*

*1) Most importantly, the reviewers felt that it would be desirable for the paper to be as accessible as possible for the general readership of* eLife*. The presentation is extremely dense in the description of the results and the biochemical figures which are hard to read because of the multiple curves.*

The reviewers stated that the Results section was very dense. We agree with this assessment and have tried to make the results text as streamlined as possible, given the significant amount of data presented in this manuscript. Additionally, we believe that the Abstract, Introduction, and Discussion sections present the work in a way that can be easily interpreted by the broader scientific community.

The reviewers stated that some of the biochemical curves were busy and difficult to interpret. We separated the panels of Figure 1into Figure 1(just panel A) and Figure 1—figure supplement1(panel B) to make the curves easier to read/interpret. We also made new figure supplements for Table 2that are easier to interpret.

*2) In addition, a model figure would make the results more accessible to the general reader.*

The reviewers stated that a model figure could provide some clarity for our mechanism. As such we created Figure 8, a model figure showing a graphical overview of our major findings.

3) Finally, the authors should discuss the implications of their work for future drug discovery and comment on how their information could predict the likely efficacy of the most contemporary SERMs/SERDs i.e. GDC910, RAD1901, AZ9694.

As the reviewers pointed out, new and improved compounds (e.g. Bazedoxifene, GDC0810, AZ9496, and RAD1901) have been reported. These compounds are now addressed in the Discussion section.